# The Effects of Heat Treatment on the Microstructure and Tensile Properties of an HPDC Marine Transmission Gearcase

**Joshua Stroh** [1,*], **Dimitry Sediako** [1] , **Ted Hanes** [2], **Kevin Anderson** [2] and **Alex Monroe** [2]

1   University of British Columbia, Okanagan 1137 University Way, Kelowna, BC V1V 1V7, Canada;
    Dimitry.Sediako@ubc.ca
2   Mercury Marine, W6250 Pioneer Road, Fond du Lac, WI 54936, USA; Ted.Hanes@mercmarine.com (T.H.);
    Kevin.Anderson@mercmarine.com (K.A.); Alex.Monroe@mercmarine.com (A.M.)
*   Correspondence: josh_stroh@alumni.ubc.ca

**Abstract:** The drive for continuously improving the performance and increasing the efficiencies of marine transportation has resulted in the development of a new alloy, Mercalloy A362™. This alloy was designed to lighten Mercury Marine's lower transmission gearcase while also improving the alloy's recyclability. The new prototype gearcase was subjected to Mercury Marine's standard service conditions, which resulted in the premature failure of the prototype. A previous study revealed that a large accumulation of unwanted residual stress (~120 MPa) was present in the gearcase following the high pressure die casting process. Fortunately, the T5 heat treatment reduced the magnitude of stress by approximately 50%. However, the effects that the T5 heat treatment had on the microstructure and mechanical properties of the alloy were not discussed. Thus, this research article characterizes the effects that the T5 heat treatment has on the volume fraction and morphology of the intermetallics, as well as the tensile performance of the alloy. It was found that the T5 heat treatment led to only minor increases in the volume fraction of Fe-bearing intermetallics, leading to similar tensile properties in both the as-cast and T5 condition. These results suggest that the T5 heat treatment can alleviate residual stress without significantly altering the mechanical properties of the alloy. The results from the previous stress analysis and the current study were used to optimize the manufacturing process which led to the successful introduction of the gearcase into the competitive marine industry.

**Keywords:** aluminum alloys; heat treatment; marine transmission gearcase; microstructure; tensile test

## 1. Introduction

The use of aluminum (Al) alloys has greatly increased over the past two decades for the production of lightweight powertrain components such as transmission gearcases and engines used in automobiles and marine crafts. The capability of producing lightweight components with complex geometries is one of the most desirable characteristics of Al powertrain alloys. For applications that operate at ambient temperatures, the hypoeutectic A356 (Al–Si–Mg-based) alloy is commonly used. Some of the reasons for its popularity include great castability, sufficient strength, and its responsiveness to chemical modification and/or heat treatment. Elements such as phosphorus (P), strontium (Sr), sodium (Na), and barium (Ba) may be added to refine the alloy's microstructure by suppressing the growth of specific phases or altering the structure of the phase into a more coherent morphology. Sr is the most common modifier because it is less reactive as compared to the other elements [1]. The addition of Sr to Al–Si alloys can refine the eutectic Si particles from coarse flakes to a more fine and fibrous structure, which improves the alloy's strength [2,3]. In addition to chemical refinement, solutionizing has been shown to fragment and spheroidize the Si particles in 0.5–4 h, depending on the state of the Al–Si alloy (i.e., modified or not) as well as solutionizing temperature [2–4]. The semi-spherical shape of the modified Si particles reduces stress concentrations and improves the alloy's yield strength (YS) and ultimate

tensile strength (UTS) [5]. A study performed by R. Chen et al. investigated the effects of quenching as well as a T6 heat treatment on the mechanical properties an Al–7 wt.%Si–Mg cast alloy [2]. Their results indicated that a 2 h solution treatment at 535 °C followed by quenching leads to an increase in the YS and UTS elongation as compared to the as-cast state. These improved properties are likely associated with the homogenization of the solute atoms, an alteration to the eutectic Si morphology, and the dissolution of weakening intermetallics. After artificial aging and completing the T6 temper, a significant increase in both the YS and UTS was observed; however, a reduction in the elongation also occurred. The increase in strength was likely attributed to the precipitation of nano-sized phases, and the decrease in elongation may be associated to the further spheroidization of eutectic Si [5].

Combating the demand for higher performance and greater efficiencies amongst the marine industry, Mercury Marine has developed a new lightweight Al alloy, Mercalloy A362™ [6], which they have used for the lower transmission gearcase on one of Mercury Marine's high-powered outboard engines. The high Si content (~11 wt.%, as compared to ~7 wt.% for A356), was expected to improve the alloy's strength and allowed for up to 33% thinner walls. Not only did the thinner walls lighten the gearcase, but the lower density also accompanied by the increased Si content further lowered the total weight of the component, as compared to its predecessor which utilized an A356 type alloy. The Mercalloy A362™ alloy was purposely developed to improve the efficiency of the marine craft (via weight reduction) and to improve the alloy's castability, machinability, and recyclability. The castability of the alloy was improved, as compared to A356, by raising the Si content closer to the eutectic Si composition, thereby providing a near-instantaneous solidification of the molten alloy [6]. The recyclability of the alloy was improved by using 100% recycled materials and limiting the individual wt.% of each solute element. Limiting the compositional requirement for the alloying elements promotes the re-melting of scrap material, thereby lowering waste, costs, and energy consumption.

Although the alloy's performance was expected to be superior to the A356 alloy, during the rigorous preliminary testing a few of the as-cast prototype gearcases failed unexpectedly due to the development of a macro crack. This was believed to be caused by the development of high residual stresses during the HPDC process. As a result, the current authors performed a comprehensive residual stress analysis using neutron diffraction on the as-cast and T5 heat-treated transmission gearcases; the results are presented elsewhere [7]. It was observed that the peak magnitude of residual stress reached nearly 120 MPa in the as-condition, accounting for a large magnitude of the material's expected strength. Fortunately, approximately half of the maximum residual stress was alleviated after applying the T5 heat treatment. Although the residual stress profiles of the gearcase were described, and it was determined that applying the T5 heat treatment would be a sufficient method for minimizing the likelihood of a crack forming, it was unclear what caused the alleviation of residual stress. Moreover, the effects from microstructural variations and/or defects were not discussed in detail. Microstructural defects such as porosity or intermetallics with harmful morphologies can act as stress concentrators, and can act as crack initiation sites [8,9], thereby decreasing the alloy's toughness and fatigue strength.

In general, the HPDC process consists of forcing molten metal into a metal die cavity by means of external pressure. After the metal solidifies, the die is opened, and the casting is ejected. Due to the relatively high applied pressure (~10–175 MPa), the casting has a near-net finish which reduces the amount of post-processing machining. Moreover, the applied pressure allows manufacturers to produce castings with relatively thin wall sections (i.e., <4 mm), which can result in the production of lightweight components. For HPDC aluminum, a steel mold is commonly used to generate greater cooling rates as compared to other casting methods such as sand casting. As a result, HPDC alloys typically have higher Fe content to minimize the likelihood of the casting sticking to the die. However, relatively high Fe concentrations usually lead to the formation of two harmful Fe-bearing intermetallics, specifically $Al_5FeSi$ and $Al_8FeMg_3Si_6$ [10]. The brittleness and

poor morphology of these Fe-bearing phases commonly leads to a decrease in the alloy's fracture toughness. To add to this, the high velocity (~20–80 m/s [11]) of molten metal that is forced into the die cavity often leads to turbulent flow, and as a result entraps gas and leads to elevated levels of porosity, further increasing stress concentrations and lowering the mechanical properties of the alloy.

Thus, to determine if microstructural defects contributed to the initiation of the crack, this research analyzed the microstructure of the new A362 alloy at several critical locations in the marine transmission gearcases. Moreover, the effects that the T5 heat treatment has on the microstructure and mechanical properties of the A362 alloy are also described.

## 2. Materials and Methods

The standard chemical composition of the A362 alloy taken from the prototype marine transmission gearcases is shown in Table 1. Samples were cut out of the as-cast and T5 heat-treated gearcase from: (i) the crack formation line, which consisted of a thin-walled section (Wall, AW, TW) and a thicker-walled section (Hub, AH, TH); (ii) from the torpedo wall (Reference, AR, TR); and (iii) from a thick shaft section (Shaft, AS, TS). Figure 1 shows the locations where each sample was taken from the gearcases.

**Table 1.** The chemical composition (wt.%) of A362 alloy taken from the prototype gearcase.

| Sample | Si | Fe | Cu | Mn | Mg | Cr | Ni | Zn | Ti | Sr | Al |
|---|---|---|---|---|---|---|---|---|---|---|---|
| A362 Gearcase | 10.8 | 0.27 | 0.07 | 0.25 | 0.58 | 0.01 | 0.01 | 0.02 | 0.08 | 0.06 | Bal. |

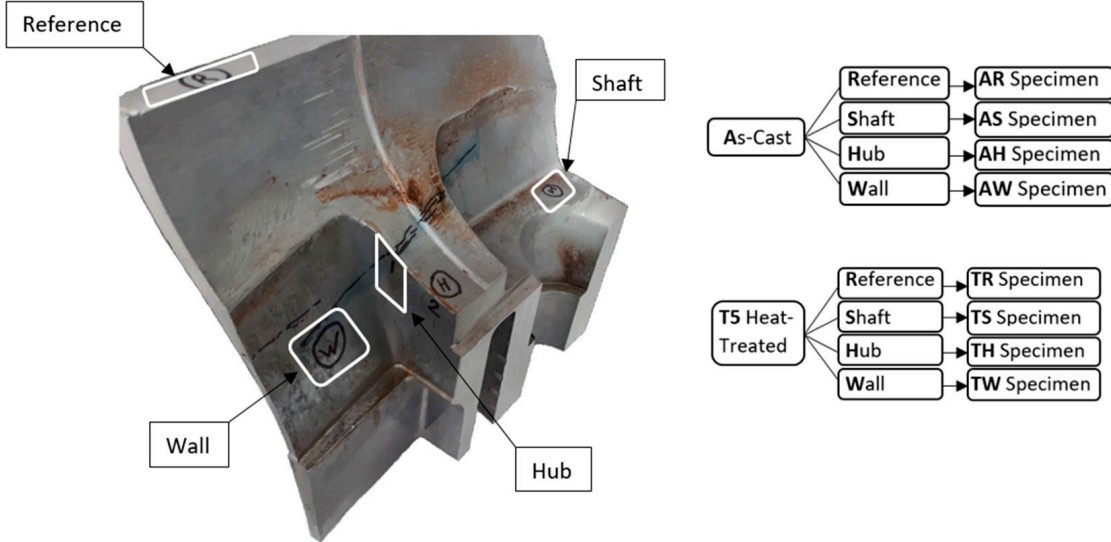

**Figure 1.** Locations and naming scheme of the samples taken from the gearcase.

Metallography samples taken from the gearcase were mounted in a two-part, fast curing acrylic resin and then sequentially ground using 240 grit to 600 grit SiC paper. The ground samples were then polished using 9, 6, 3, and 1 μm diamond paste. Finally, each sample was electroetched at 30 V for 15 s in Barker's etchant (5 mL fluoroboric acid in 200 mL distilled water). A stainless-steel cathode was used.

A Zeiss optical stereoscopic microscope (OSM) was used to measure the grain size and secondary dendrite arm spacing (SDAS) with an accuracy of ±0.1 μm. The SDAS was measured to determine the cooling rate at each location during the casting process. The cooling rate (*CR*) was calculated using Equation (1) [12]:

$$SDAS = 36.1(CR)^{-0.34} \tag{1}$$

The OSM and a field emission gun scanning electron microscope (FEG-SEM, Tescan Mira3 XMU, Kohoutovice, Czech Republic) operating at an accelerating voltage of 20 kV were used to characterize the microstructure. Oxford Instruments Aztec data acquisition and processing software (version 2.4, Bristol, UK) with an 80 mm$^2$ Oxford EDS detector (Bristol, UK) were utilized for the composition analysis of the various phases. Morphology and EDS spectra for each phase was compared to the literature and the inorganic crystal structure database (ICSD) [13] to determine the approximate stoichiometry of each phase.

To determine how the microstructure and likely differed in cooling rates between the varying wall thickness locations, tensile tests were performed on the Reference (AR/TR), Shaft (AS/TS), and Hub (AH/TH) sections of the marine gearcase (see Figure 1). Dog bone-shaped tensile samples were first extracted from the gearcases with a waterjet and then milled to the final dimensions, as outlined in the ASTM standards E8 M [14].

A 25 mm MTS extensometer (MTS, Eden Prairie, MN, USA) was used to measure the axial strain that each sample experienced while the MTS Landmark Servohydraulic system pulled the samples at a constant strain rate of 0.2 mm/mm/min. The results were processed to determine the ultimate tensile strength (UTS), yield strength (YS) at 0.2% strain, Young's modulus, and the elongation.

## 3. Results and Discussion

### 3.1. Microstructure

The results discussed below are for the samples taken from the as-cast gearcase. The average SDAS and calculated cooling rate for the reference (AR), thick shaft (AS), and crack location (AW and AH) samples are summarized in Table 2. As expected, the thinnest walled section (AW) had the smallest SDAS and thus the fastest cooling rate, reaching ~14.1 °C/s. Correspondingly, the thickest section experienced the slowest cooling, reaching only 7.8 °C/s. Such a large difference in cooling rates was described to be one of the causes for the elevated level of residual stress [7] and is expected to result in somewhat dissimilar microstructures and thus mechanical properties.

**Table 2.** Summary of the grain size, secondary dendrite arm spacing (SDAS) and cooling rate for the as-cast component.

| Measurement | AH | AR | AS | AW |
|---|---|---|---|---|
| SDAS ± SD (μm) | 18 ± 2 | 16 ± 2 | 16 ± 2 | 15 ± 3 |
| Cooling Rate (°C/s) | 7.8 | 11.7 | 11.8 | 14.1 |

The microstructure of the AH, AR, AS, and AW specimens are shown in Figures 2–5. The micrographs were selected to purposely show the intermetallics, and thus the bulk of the microstructure at each location may vary slightly; however, the volume fraction measurements listed in Table 3 are correct. The darker grey, continuous phase as shown in Figures 2–5 was observed in all samples and typically occupied the majority of the samples' surface area. The morphology and composition of this phase indicates that it is the primary $\alpha$-Al phase, which is consistent with the literature on similar alloy compositions [15–17]. The laminated structure (alternating dark and light grey phase) was also observed in all samples. The morphology and composition of this phase indicates that it is the Al–Si eutectic phase. Previous findings in the literature mention that this phase has various morphologies and compositions depending on the alloying elements present [15,18,19]. The blocky, fibrous structure of the Si particles corresponds well with the as-cast microstructure of Al–Si alloys.

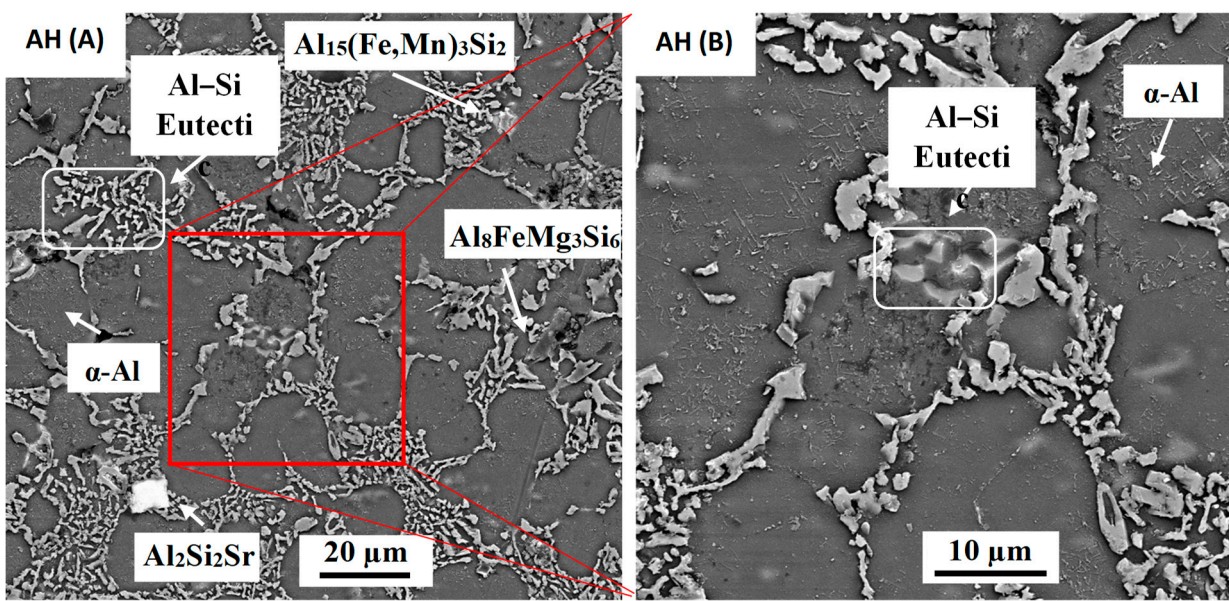

**Figure 2.** Representative micrographs for the as-cast component: (**A**) AH location; (**B**) close-up image of the red box region in (**A**) showing various phases.

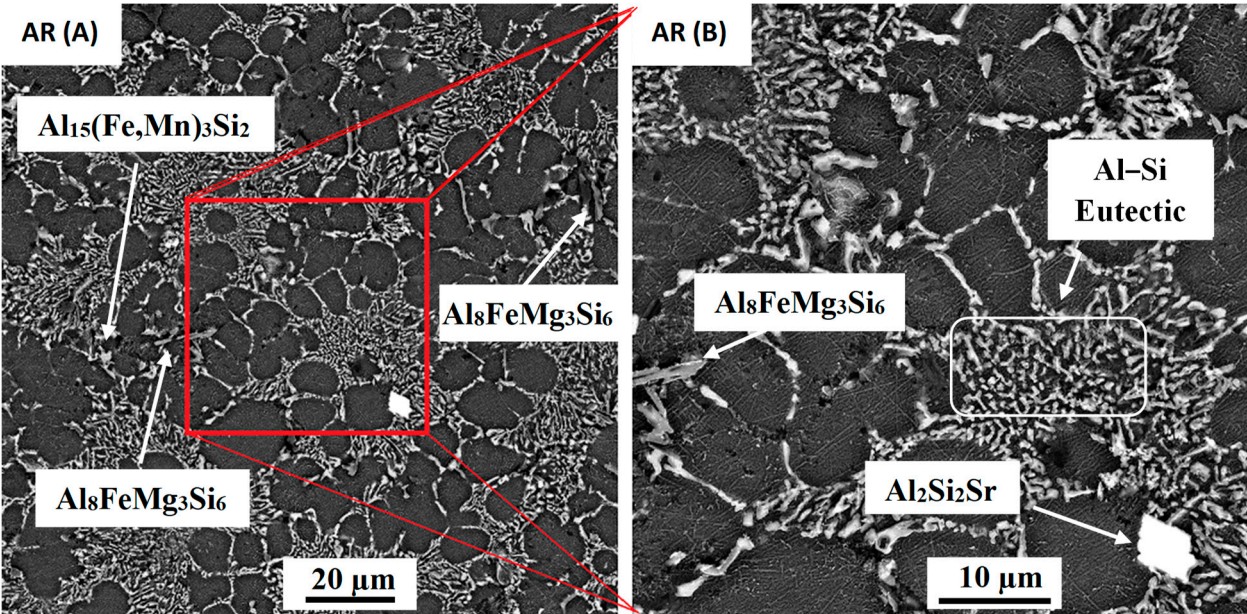

**Figure 3.** Representative micrographs for the as-cast component: (**A**) AR location; (**B**) close-up image of the red box region in (**A**) showing various phases.

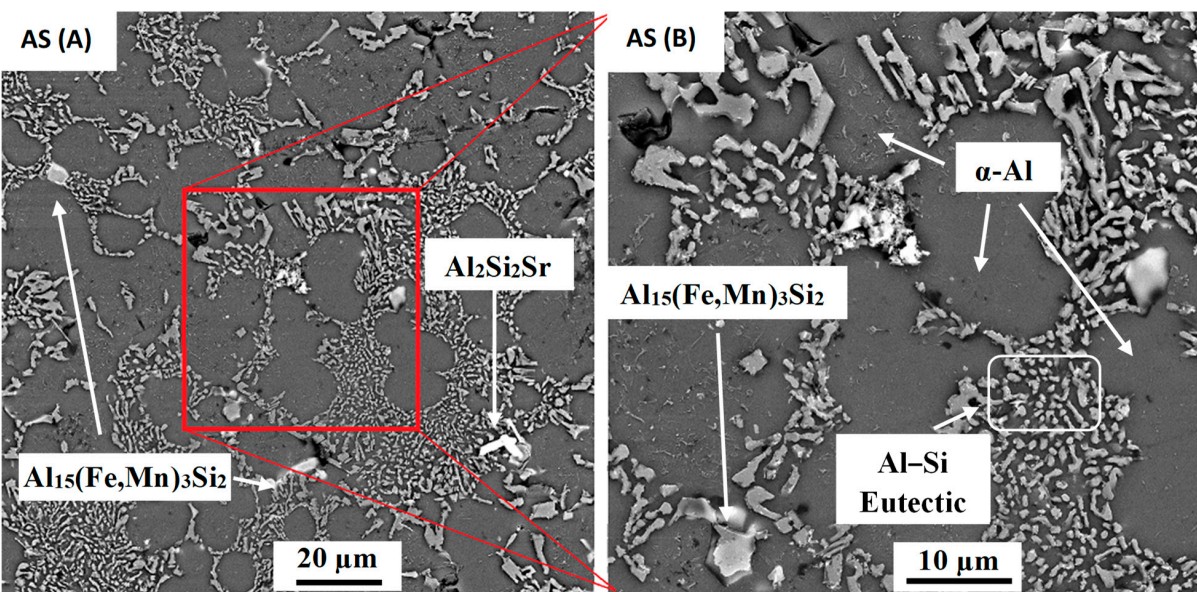

**Figure 4.** Representative micrographs for the as-cast component: (**A**) AS location; (**B**) close-up image of the red box region in (**A**) showing various phases.

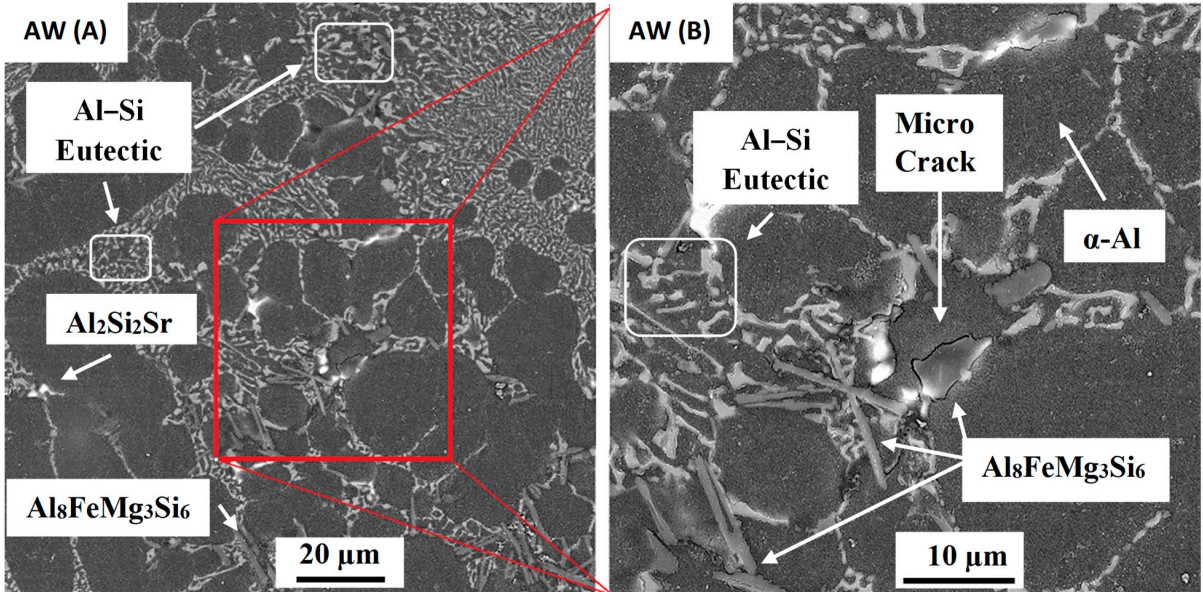

**Figure 5.** Micrographs for the as-cast component showing: (**A**) AW location; (**B**) close-up of the red box region in (**A**) showing various phases.

**Table 3.** Summary of the average volume fraction for various phases of interest in the as-cast component.

| Phases | As-Cast A362 Gearcase (Vol.% $\pm$ 95% C.I.) | | | |
|---|---|---|---|---|
| | **AH** | **AR** | **AS** | **AW** |
| $\alpha$-Al | Bal. | Bal. | Bal. | Bal. |
| Al–Si eutectic | $53 \pm 3$ | $48 \pm 3$ | $48 \pm 4$ | $55 \pm 4$ |
| $Al_8FeMg_3Si_6$ | $2.0 \pm 0.5$ | $2.3 \pm 0.9$ | $1.0 \pm 0.2$ | $3.0 \pm 0.5$ |
| $Al_{15}(Mn,Fe)_3Si_2$ | $2.0 \pm 0.5$ | $1.5 \pm 0.4$ | $1.7 \pm 0.2$ | $2.3 \pm 0.2$ |
| $Al_2Si_2Sr$ | $1.0 \pm 0.3$ | $1.5 \pm 0.6$ | $1.5 \pm 0.5$ | $1.3 \pm 0.3$ |
| Porosity | $1.9 \pm 0.2$ | $0.6 \pm 0.1$ | $0.9 \pm 0.1$ | $0.4 \pm 0.1$ |

In addition to primary Al and eutectic Si, the AH sample had a small volume fraction of blocky $Al_2Si_2Sr$ (~1.0 vol.%), needle-like $Al_8FeMg_3Si_6$ (2.0 vol.%), and blocky $Al_{15}(Mn,Fe)_3Si_2$ (2.0 vol.%). The needle-like morphology of the $Al_8FeMg_3Si_6$ acts as a stress concentrator, and the brittleness of the phase lowers the alloy's toughness. It is common for manufacturers to add manganese (Mn) to the alloy to purposely transform this phase into the more favorable $Al_{15}(Mn,Fe)_3Si_2$ phase. The blocky/polygonal morphology of $Al_{15}(Mn,Fe)_3Si_2$ lowers stress concentrations as compared to the needle-like morphology, and as a result this phase improves the alloy's toughness [10,20,21]. The slow cooling rate that the AH section experienced resulted in a considerable amount of porosity (~1.9 vol.%). Although efforts are placed on minimizing the development of porosity characteristics, in practical terms, about 1 vol.% is generally acceptable in industry.

The faster-cooling AR sample resulted in a considerably lower volume fraction of porosity as compared to the AH sample (i.e., 0.9 vs. 1.9 vol.%). However, a higher volume fraction of the harmful $Al_8FeMg_3Si_6$ (2.3 vol.%) phase was present. Although harmful, it is more likely that porosity would cause crack initiation and thus the benefits associate with the 50% reduction in porosity should outweigh the stress-concentrating effects from the $Al_8FeMg_3Si_6$ phase.

The microstructures at the AS and AR locations were very similar. The primary difference between the two locations was a slight increase in the porosity at AS. Combating the negative effects associated with the increased volume fraction of porosity, it was found that a much lower volume fraction of the $Al_8FeMg_3Si_6$ (i.e., 1.0 vs. 2.3 vol.%) phase was present. The lack of correlation between cooling rates and the volume fraction of $Al_8FeMg_3Si_6$ suggests that it may be more correlated with the casting process instead. A similar amount of $Al_{15}(Mn,Fe)_3Si_2$ was observed in the AS sample as compared to the AR sample.

Although the location with the thinnest wall thickness, and thus, fastest cooling rate contained the lowest volume fraction of porosity (i.e., 0.4%), the volume fraction of the needle-like $Al_8FeMg_3Si_6$ phase was the highest of all four locations. Moreover, the size of this phase was notably larger in AW as compared to all other locations (see Figure 5). Additionally, some micro-cracks were observed in the AW specimen, as shown in Figure 5B. These cracks were found along the $Al_8FeMg_3Si_6$ phase and may have formed during solidification of the alloy or during the operation testing of the cast gearcase [22].

The ASTM Standard E562-02 [23] was used for statistically estimating the volume fraction of the identifiable phases. The volume fraction is determined assuming that the area fraction is equivalent to the volume fraction for the randomly distributed particles [23,24]. The volume fraction of the phases of interest and the observed porosities in the as-cast component are summarized in Table 3. Representative micrographs of un-etched samples showing the porosities within each region are shown in Figure 6.

As seen in Table 3 and Figure 6, the thick section in the crack area (AH specimen) had the largest amount of porosity over any of the other sections. In addition, the sample with the highest cooling rate, AW, showed the lowest amount of porosity. Studies performed by Fahad et al. [25] and Michalik et al. [26] indicated that both the porosity size and volume fraction increase with a decreasing cooling rate. This correlates well with the trend found in the current study. Porosity can act as a localized stress concentrator and can significantly reduce the mechanical properties of materials [8,9], and thus it is expected that the tensile performance of the AH sample will be the lowest of all four samples.

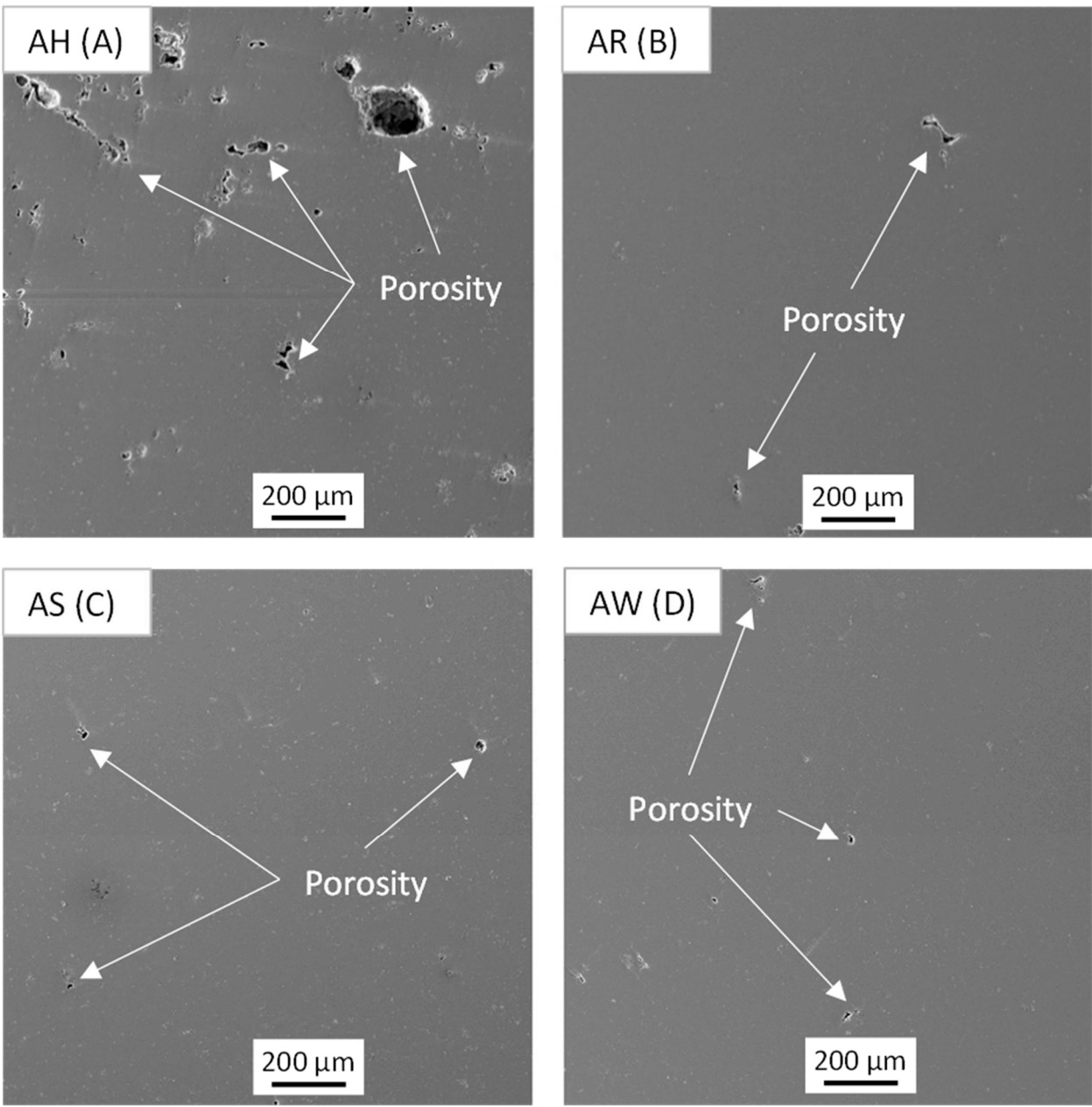

**Figure 6.** Representative un-etched micrographs for various regions of the as-cast component showing the presence of porosities: (**A**) AH specimen; (**B**) AR specimen; (**C**) AS specimen; and (**D**) AW specimen.

### 3.2. Mercalloy A362 T5 Heat-Treated

Table 4 presents the SDAS measurements and calculated cooling rates for all locations in the T5 heat-treated gearcase. The difference between the SDAS in the as-cast and in the heat-treated component is negligible because each value lies within the range of the respective standard deviations. Moreover, the trend for the T5 treated gearcase is very similar to the as-cast gearcase (i.e., H samples have the largest SDAS, and W samples have the smallest SDAS), which is expected because the heat treatment process should have no effect on the alloy's SDAS.

**Table 4.** Summary of the grain size, SDAS and cooling rate for the T5 heat-treated component.

| Measurement | TH | TR | TS | TW |
|---|---|---|---|---|
| SDAS $\pm$ SD ($\mu$m) | $17 \pm 3$ | $15 \pm 4$ | $16 \pm 4$ | $14 \pm 4$ |
| Cooling Rate ($^\circ$C/s) | 9.2 | 12.5 | 11.1 | 17.3 |

SEM micrographs for the TH, TR, TS, and TW specimens are shown below in Figures 7–10. It was observed that the T5 heat treatment had a negligible effect on the amount, size, or shape of the eutectic Si particles. However, it was observed that the morphology of the $Al_{15}(Fe,Mn)_3Si_2$ phase changed slightly from a blocky shape to a hexagonal feature after applying the T5 heat treatment process. In addition, the $Al_{15}(Fe,Mn)_3Si_2$ phase was observed in amounts of 2.3–3.4 vol.% in the heat-treated samples, slightly greater than that observed for the as-cast gearcase (i.e., 1.5–2.3%). Table 5 shows the measured volume fractions for each of the intermetallics in the T5 gearcase samples. The precipitate ($Al_{15}(Fe,Mn)_3Si_2$ phase) has been shown in the literature to have various morphologies, including flower-like [27], needle-like [28], rhombic dodecahedron [29], Chinese script [27], blocky, and polyhedral structures [30], depending on the cooling rate, heat-treatment, and the ratio of Fe to Mn. Gao et al. [27] reported that the flower-like structured $Al_{15}(Fe,Mn)_3Si_2$ phase significantly improved the tensile strength of the Al–6Si–2Fe–xMn alloys.

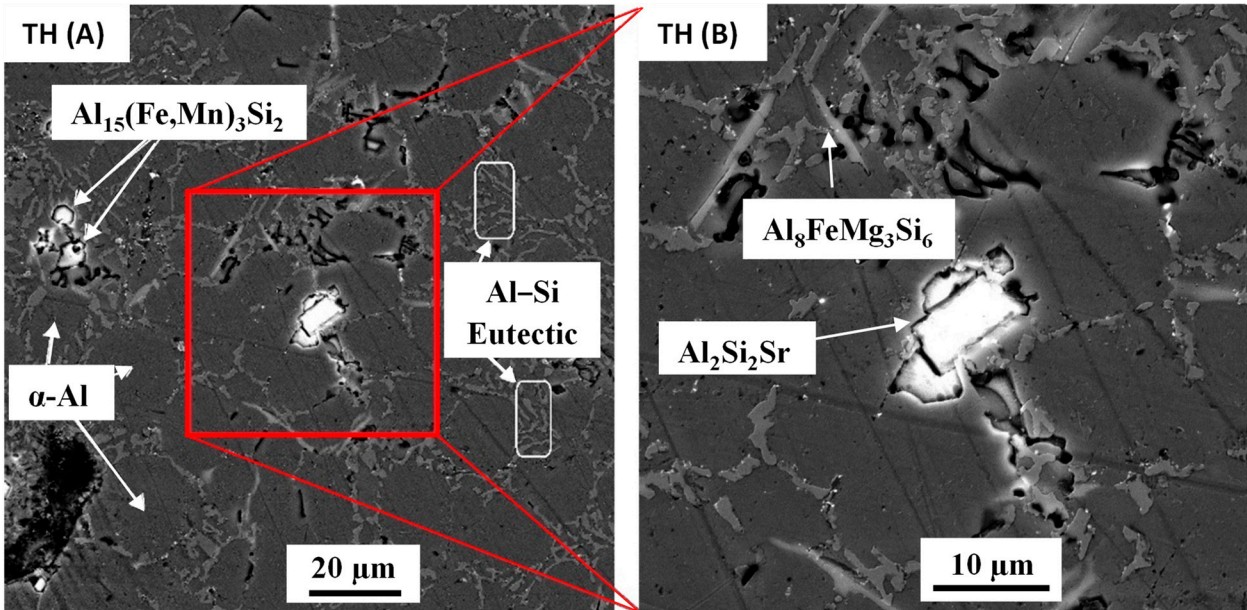

**Figure 7.** Micrograph showing the TH specimen: (**A**) representative primary phases; (**B**) close-up of the red box region in (**A**).

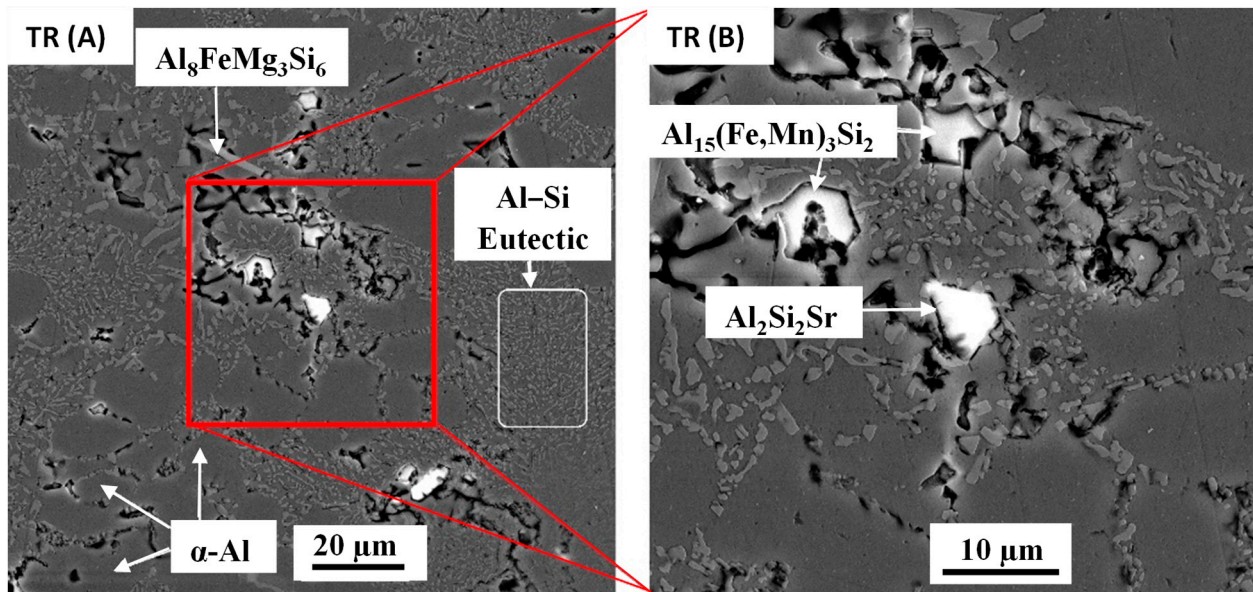

**Figure 8.** Micrograph showing the TR specimen: (**A**) representative primary phases; (**B**) close-up of the red box region in (**A**).

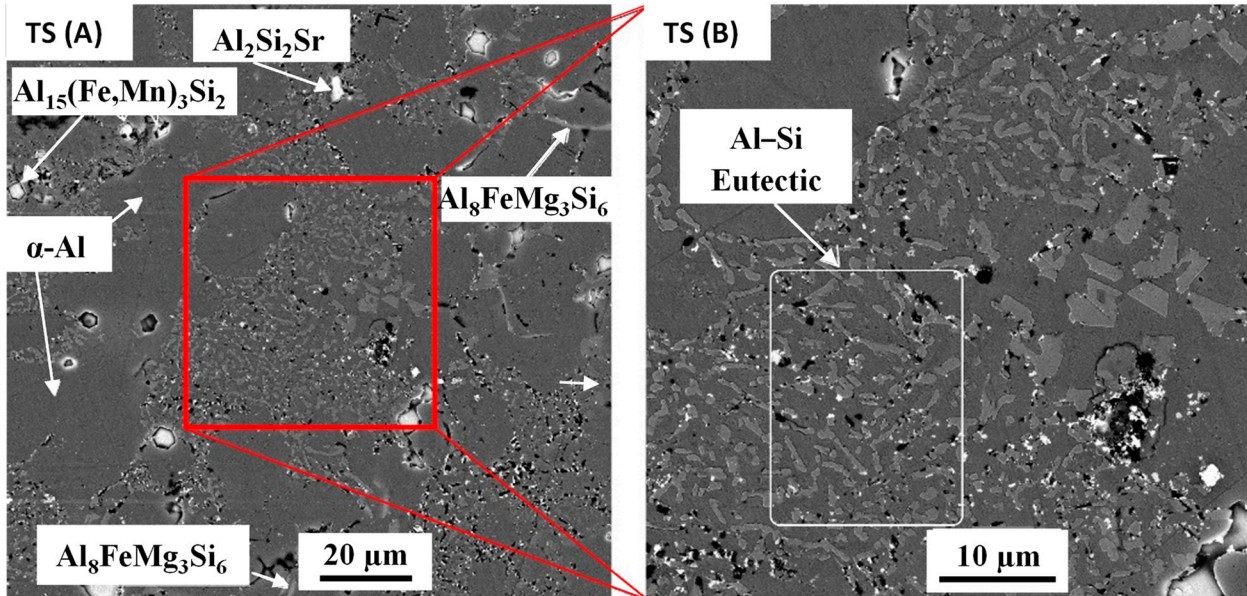

**Figure 9.** Micrograph showing the TS specimen: (**A**) representative primary phases; (**B**) close-up of the red box region in (**A**).

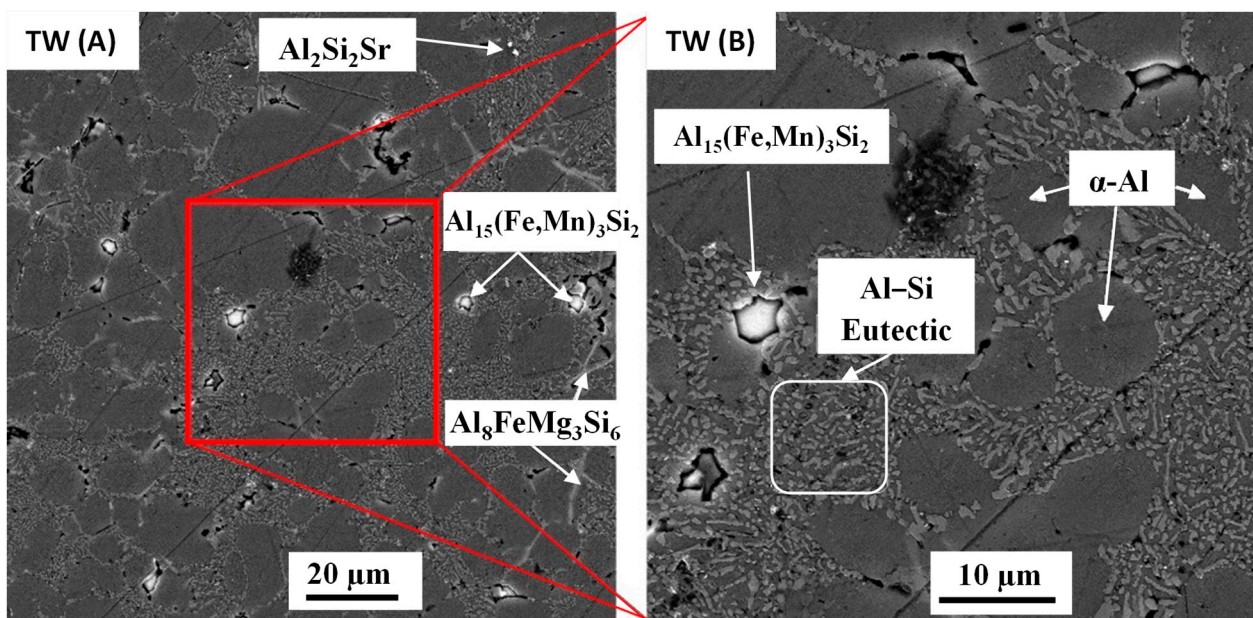

**Figure 10.** Micrograph showing TW specimen: (**A**) representative primary phases; (**B**) close-up of the red box region in (**A**).

**Table 5.** Summary of the average volume fraction for the porosities and various phases of interest in the T5 heat-treated component.

| Phases | As-Cast A362 Alloy (Vol.% $\pm$ 95% C.I.) | | | |
|---|---|---|---|---|
| | **TH** | **TR** | **TS** | **TW** |
| $\alpha$-Al | $43 \pm 4$ | $42 \pm 3$ | $46 \pm 3$ | $39 \pm 2$ |
| Al–Si eutectic | $47 \pm 2$ | $46 \pm 3$ | $43 \pm 5$ | $51 \pm 3$ |
| $Al_8FeMg_3Si_6$ | $2.1 \pm 0.4$ | $1.9 \pm 0.3$ | $2.5 \pm 0.6$ | $3.6 \pm 0.4$ |
| $Al_{15}(Fe,Mn)_3Si_2$ | $3.4 \pm 0.6$ | $2.3 \pm 0.3$ | $3.2 \pm 0.4$ | $3.4 \pm 0.3$ |
| $Al_2Si_2Sr$ | $1.1 \pm 0.2$ | $0.9 \pm 0.2$ | $1.3 \pm 0.2$ | $1.4 \pm 0.3$ |
| Porosity | $1.6 \pm 0.3$ | $0.8 \pm 0.2$ | $1.3 \pm 0.3$ | $0.7 \pm 0.2$ |

Similar to the as-cast microstructure, the $Al_2Si_2Sr$ phase was also observed. The literature indicates that this phase has a tendency to be very brittle [31] and can lower the material's toughness. Moreover, the harmful, needle-like $Al_8FeMg_3Si_6$ phase was also observed in each of the samples but was present in a slightly higher amount. The rise in the amount of precipitate is expected to improve the alloys performance during the tensile test, via the Orowan loop effect (i.e., the greater the number of small precipitates, the more difficult it is for dislocations to travel through the material). However, the brittleness and stress concentrating effect from the $Al_8FeMg_3Si_6$ phase lowers the alloys' ability to absorb crack energy.

The volume fraction of the Al–Si eutectic and intermetallic phases in the heat-treated TW specimen was slightly higher than that of the other heat-treated specimens. This is likely due to the higher in-casting cooling rate in the TW specimen as compared to the other T5 specimens. The SEM images in Figure 11 demonstrate the presence of porosities in all of the heat-treated specimens. Comparing the SDAS of the as-cast and the T5 treated samples, the SDAS values were relatively similar for all locations. Therefore, the microstructural differences (i.e., phases present, phase growth and stability) of the T5 samples in comparison to the as cast samples were likely caused solely by the T5 treatment.

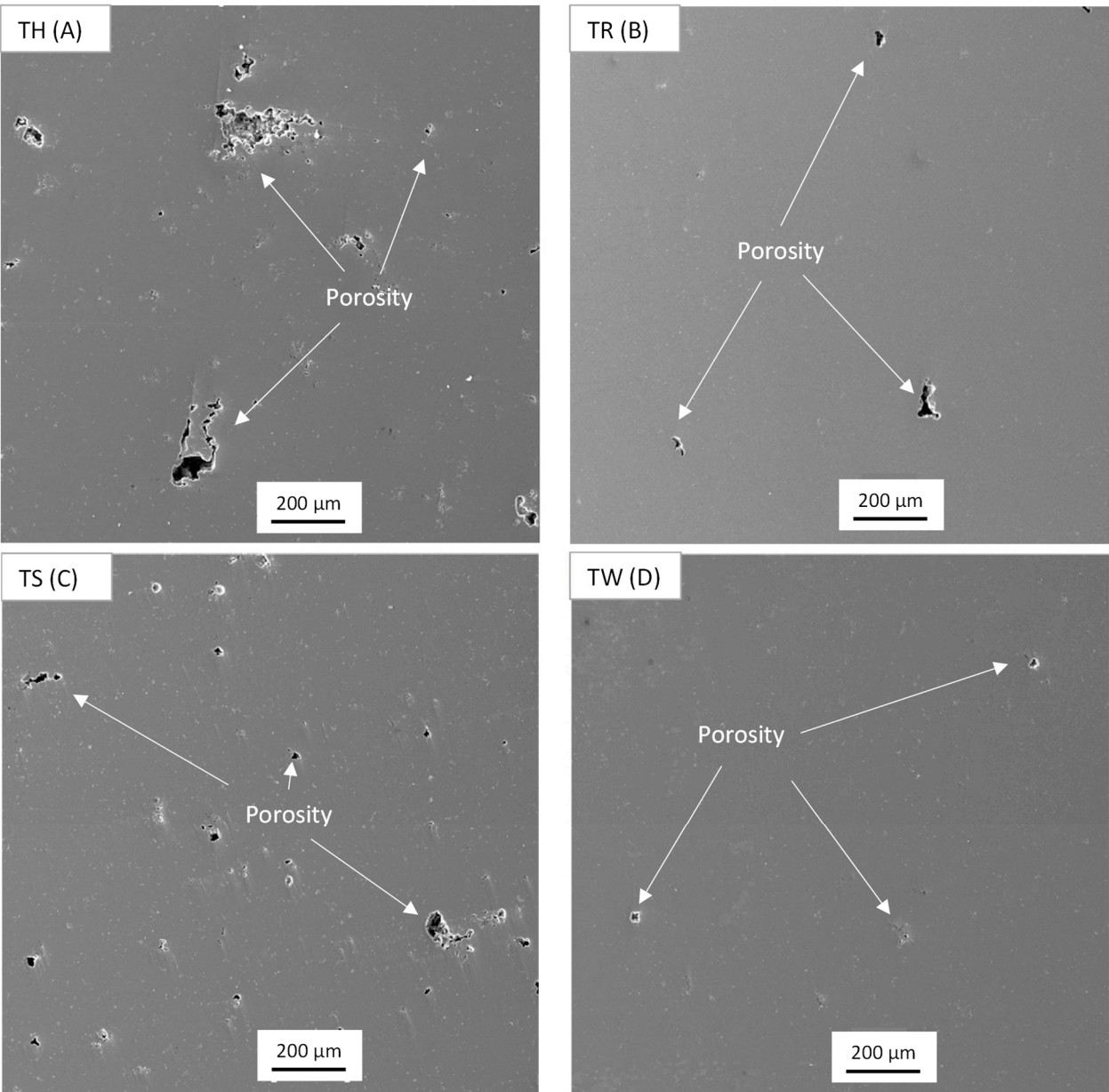

**Figure 11.** Representative un-etched micrographs for various regions of the T5 component showing the presence of porosities: (**A**) TH specimen; (**B**) TR specimen; (**C**) TS specimen; and (**D**) TW specimen.

As summarized in Table 5 and shown in Figure 11, the amount of porosity in the thick section of the crack area (TH specimen) was higher than that of the other sections. This is likely a result of TW having the slowest cooling rate as compared to the other locations [25,26]. Increasing the cooling rate during solidification has been shown to reduce the formation of porosities in the material. This is due to the reduced solidification time, and therefore there is less time for dissolved gases to be excreted from the molten metal. The size and shape of the porosities are two important factors affecting the mechanical properties of the material. Porosities found in the as-cast and T5 heat-treated material could be gas porosities or shrinkage porosities which are formed during the casting process. Porosities in the cast material typically act as stress concentrating points, frequently causing the initiation of microcracks. The larger and greater the number of porosities, the lower the fatigue life of the material due to the facilitated propagation of cracks through a porous metal.

As shown in Figure 12, porosity can be seen forming around the $Al_2Si_2Sr$ phase. It has been reported that elevated levels of Sr can lead to the increased formation of Sr-containing phases, such as $Al_2Si_2Sr$, which can promote the development of porosity in the material [32].

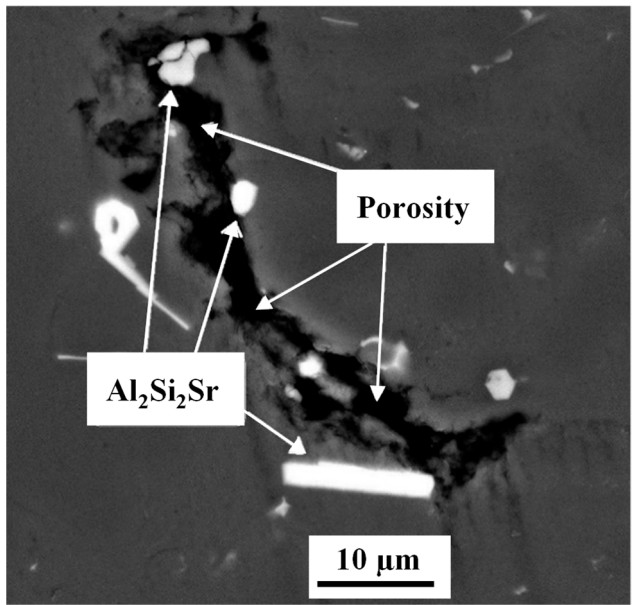

**Figure 12.** SEM micrograph of TS showing the formation of porosity near the $Al_2Si_2Sr$ phase.

## 4. Tensile Properties

A summary of the tensile test results for the as-cast gearcase is presented below in Table 6. Due to the complex curvature and small wall thickness of the Wall location (AW/TW), tensile samples could not be extracted from this location.

**Table 6.** Tensile test results for as-cast gearcase.

| Properties | AR | AS | AH |
|---|---|---|---|
| UTS (MPa) | $205 \pm 10$ | $178 \pm 10$ | $200 \pm 10$ |
| YS (MPa) | $167 \pm 2$ | - | $163 \pm 2$ |
| Young's Modulus (GPa) | $72.4 \pm 1.5$ | $68.9 \pm 1.5$ | $69.6 \pm 1.5$ |
| Percent elongation (%) | $1.0 \pm 0.2$ | $0.4 \pm 0.2$ | $0.9 \pm 0.2$ |

The AR location had the greatest mechanical properties as compared to the other locations. This is likely due the faster cooling rate, leading to smaller grain sizes and the lowest volume fraction of porosity (see Table 3). The AS sample had visible porosity after machining, which effectively lowers the cross-sectional area and is believed to be the primary cause for the poor tensile performance. It should be noted that the AS sample broke before 0.2% strain was achieved, and thus the YS could not be obtained. The engineering stress–strain curves for each sample are shown below in Figure 13.

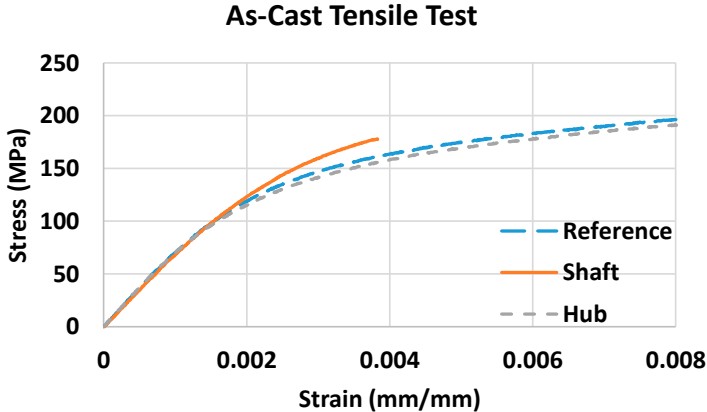

**Figure 13.** As-cast tensile test results.

Compared to the as-cast tensile properties of an A356 alloy [33,34], the A362 alloy had a much higher YS (163–167 vs. 97–104 MPa) but a considerably lower UTS (178–205 vs. 240 MPa). This is likely attributed to the higher volume fraction of porosity in HPDC A362 samples. Porosity has less of an effect on the YS of Al alloys but has a much more pronounced effect on the UTS and elongation [35]. In addition to porosity, the relatively high volume fraction of needle-like $Al_8FeMg_3Si_6$ precipitates (see Figures 2–5 and Table 3) promotes stress concentrations, and the high aspect ratio increases the interfacial energy between the precipitate and the matrix. As a result, the phase acts a crack initiation site and lowers the alloy's ductility [36]. Moreover, the formation of the $Al_8FeMg_3Si_6$ phase consumes a large portion of the available Mg for forming the more desirable intermetallic $Mg_2Si$, which is the predominant phase that improves the strength of Al–Si–Mg alloys following heat treatment [37,38].

The tensile test results for the T5 heat-treated gearcase are shown below in Table 7, and the engineering stress–strain curves are displayed in Figure 14. The results follow a similar trend to the as-cast samples. Compared with the as-cast samples, the TR location had a slightly improved YS and elongation, following by TH, and finally TS which had the poorest performance.

**Table 7.** Tensile test results for the T5 heat-treated gearcase.

| Properties | Reference | Shaft | Hub |
|---|---|---|---|
| UTS (MPa) | $220 \pm 10$ | $169 \pm 10$ | $200 \pm 10$ |
| YS (MPa) | $169 \pm 2$ | $161 \pm 2$ | $160 \pm 2$ |
| Young's Modulus (GPa) | $71.1 \pm 1.9$ | $66.8 \pm 1.9$ | $73.4 \pm 1.9$ |
| Percent elongation (%) | $1.3 \pm 0.3$ | $0.5 \pm 0.3$ | $1.0 \pm 0.3$ |

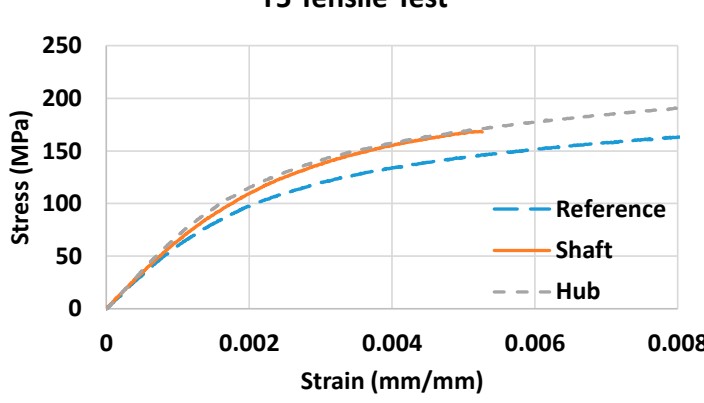

**Figure 14.** T5 heat-treated tensile test results.

Similar to the as-cast gearcase, the quicker cooling and lower volume fraction of porosity is believed to be the primary causes for the improved tensile performance of the TR sample, as compared the TS and TH specimen. Moreover, the presence of shrinkage porosity near the $Al_2Si_2Sr$ intermetallics (see Figure 12) in the TS sample is also believed to be one of the reasons for the impaired performance of this specimen.

The similar microstructures and mechanical properties between the as-cast and T5 A362 alloy suggest that Mercury Marine's specific heat treatment did not lead to any property improvements via modification of the intermetallics. It is likely that that the solutionizing temperature and time are lower than those required to modify the microstructure. Similar to the results presented by P. Cavalier et al. and L. BoChao et al. [33,34], the YS and UTS of an A356 alloy were insignificantly affected by a 160–200 °C aging process until the time was increased well above 2–4 h (which is economically unfeasible for mass production). In addition, the relatively high volume fraction of $Al_8FeMg_3Si_6$ precipitates in the A362 alloy consumed a large portion of the available Mg for forming $Mg_2Si$ during the T5 heat treatment [37,38]. The lack of $Mg_2Si$ in the microstructure of the T5–A362 samples supports this claim. This is believed to be the primary reason for the similar tensile properties between the as-cast and T5 heat-treated samples.

The lack of microstructural changes following the T5 heat treatment suggests that the T5 heat treatment likely alleviates the stress via plastic deformation caused by the residual stress surpassing the material's temperature sensitive yield strength during the solutionizing and subsequent aging portion of the T5 heat treatment.

## 5. Conclusions

This study investigated the effects that a commercial T5 heat treatment has on the microstructure and tensile performance of a high-pressure die-cast prototype marine transmission gearcase. The conclusions from the present study are described as follows:

(1). The T5 heat treatment has a negligible effect on the grain size, SDAS, and the amount/size/shape of the eutectic Si particles. However, the heat treatment led to a slightly greater volume fraction of intermetallics, specifically $Al_8FeMg_3Si_6$ and $Al_{15}(Mn,Fe)_3Si_2$. The harmful, needle-like morphology of the $Al_8FeMg_3Si_6$ induces stress concentrations and its brittleness can further deteriorate the mechanical properties. By transforming the needle-like $Al_8FeMg_3Si_6$ phase into the blocky/polygonal $Al_{15}(Mn,Fe)_3Si_2$ compound, through the addition of Mn or modified cooling rates, the stress concentrations in the material decrease, and as a result, the tensile performance generally improves. Although some microstructural changes were observed, the differences between the as-cast and T5 gearcase are minor and thus are not presumed to be correlated to the alleviation of residual stress described the previous stress analysis [7];

(2). Due to the similar microstructures between the as-cast and T5 gearcases, the tensile performance of the samples taken from each component were similar. The T5 reference sample showed slightly greater properties compared to the as-cast reference sample, which could be correlated to the small increase in $Al_{15}(Mn,Fe)_3Si_2$ and decrease in $Al_8FeMg_3Si_6$; however, this trend is not consistent for the other samples;

(3). The near-equal tensile performance and similar microstructures observed for both gearcases suggests that the alleviation of residual stress, that was observed in the preliminary stress analysis [7], following heat treatment was primarily caused by temperature-induced plasticity that occurred in the material during the elevated temperature solutionizing or precipitation stage of the heat treatment. It is likely that the high magnitude of residual stress (~120 MPa) that was present in the gearcase exceeded the temperature-dependent yield strength of the A362 alloy, leading to a stress release via plasticity.

The combined results from this microstructure and tensile performance study, as well as the residual stress analysis, suggest that the elevated magnitudes of residual stress and presence of relatively high levels of porosity in the as-cast gearcase are two of the primary

causes for the initially observed macrocrack. Moreover, it was determined that depending on several operational factors (i.e., engine speed, propeller diameter/pitch/material, water inflow velocity), the water may cavitate around the propeller, leading to a high-magnitude torque fluctuation at frequencies of 45–60 Hz. This frequency happened to coincide with the first resonant frequency of the drivetrain. As a result, the dynamic torque experienced by the drivetrain increased considerably, and combined with the residual stress, surpassed the material's yield strength. It is likely that the porosity first acted as a crack initiation site and then the high magnitude of residual stress and cyclic operational loading led to the rapid propagation of the crack. However, the in-tank test conditions were different from the actual operational conditions which the gearbox would experience when mounted to a service-ready marine craft. The increased influx velocities eliminate or minimize cavitation, and the cavitation-induced torque fluctuations are not of concern during actual in-service conditions.

Moreover, the analysis of the T5 gearcase revealed that the T5 treatment is capable of alleviating a considerable amount of residual stress without affecting the alloy's strength. As a result, the combined stresses in the T5 gearcase are low enough to prevent crack formation, and thus, no design or manufacturing alterations were required. The T5 gearcase has been successfully introduced to the marine industry and is showing little to no signs of failure.

**Author Contributions:** Conceptualization, D.S. and T.H.; methodology, J.S. and D.S.; validation, J.S. and D.S.; formal analysis, J.S. and D.S.; investigation, J.S., D.S., T.H., K.A. and A.M.; resources, D.S., T.H., K.A. and A.M.; data curation, J.S. and D.S.; writing—original draft preparation, J.S.; writing—review and editing, J.S., D.S., T.H., K.A. and A.M.; visualization, J.S. and D.S.; supervision, D.S.; project administration, T.H.; funding acquisition, D.S. and T.H. All authors have read and agreed to the published version of the manuscript.

**Funding:** This research received no external funding.

**Institutional Review Board Statement:** Not applicable.

**Informed Consent Statement:** Not applicable.

**Data Availability Statement:** The raw data required to reproduce the tensile results are available to download from Mendeley Data ["Data for Mercury Marine Gearcase", http://dx.doi.org/10.17632/mwkt65b7jn.1] (accessed on 28 February 2021).

**Conflicts of Interest:** The authors declare no conflict of interest.

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
