# Peer review of "The Effects of Heat Treatment on the Microstructure and Tensile Properties of an HPDC Marine Transmission Gearcase"

_metals, doi:10.3390/met11030517_

Round 1

Reviewer 1 Report

  1. From line 71 to line 73, the definitions of these symbols AW, AH, AR, AS, TW, TH, TR, TS should be consistent with those shown in Figure 1.
  2. "Table 1" described in line 103 to line 109 is reused and must be corrected. All the following table numbers must be renumbered correctly.
  3. On line 111, "Figure 2 to Table 5" seems to be a wrong statement.
  4. In Figure 2, the locations of Al2Si2Sr, Al8FeMg3Si6 and Al15(Mn,Fe)3Si2 described in lines 121 and 122 should be marked.
  5. Description of "A similar amount of Al15(Mn,Fe)3Si2 was observed in the AS sample as compared to the AR sample." was expressed in line 148 and line 149. However, Figure 3 showing the AR sample does not specify which position is Al15(Mn,Fe)3Si2; the associated Table 2 (actually Table 3) is not mentioned until the description on line 166. It is hard for readers to fully understand what is expressed.
  6. In Table 4 (actually Table 5) on line 266, "A362 As-Cast Component" seems to be wrong.
  7. In the article, the description of Figures 7 to 10 refers to "Al15(Fe,Mn)3Si2 phase", but Figures 7 to 10 show "Al(Mn,Fe)Si phase". The figure and text should be revised to be consistent.
  8. The doubt as in question 1 appears again; in Table 5 on line 263 (actually Table 6) and in Table 6 on line 290 (actually Table 7), the item name "Fin" is different with the "Wall" shown in Figure 1. A uniform name should be used for the specified objects throughout the article.
  9. In the tensile test results of the case as-cast (observed in Table 5), the UTS of the other three items except the Fin item are lower than the UTS of MercAlloy™ A362; and the YS of the other three items except the Shaft item are all higher than the YS of MercAlloy™ A362. On the other hand, in the T5 heat-treated tensile test results (observed in Table 6), whether it is UTS or YS, the values ​​of all items (Reference, Shaft, Hub, Fin) are lower than the values ​​of MercAlloy™ A362. As for the reasons for the different results of these two cases, the author should provide more descriptions.

10. The reference number must be revised.

Author Response

Reviewer 1

Thank you for you comments and suggestions. We have done our best to address them all and we believe the paper has been improved because of these changes.

1. From line 71 to line 73, the definitions of these symbols AW, AH, AR, AS, TW, TH, TR, TS should be consistent with those shown in Figure 1.

Response to 1: Addressed. The Reference, wall, hub, shaft definitiions have been included in the earlier described paragraph.

2. "Table 1" described in line 103 to line 109 is reused and must be corrected. All the following table numbers must be renumbered correctly.

Response to 2: Addressed and corrected.

3. On line 111, "Figure 2 to Table 5" seems to be a wrong statement.

Response to 3: You are correct, it was a typo and should be Figure 2 to Figure 5.

4. In Figure 2, the locations of Al2Si2Sr, Al8FeMg3Si6 and Al15(Mn,Fe)3Si2 described in lines 121 and 122 should be marked.

Response to 4: Addressed. Phases have been added to figure

5. Description of "A similar amount of Al15(Mn,Fe)3Si2 was observed in the AS sample as compared to the AR sample." was expressed in line 148 and line 149. However, Figure 3 showing the AR sample does not specify which position is Al15(Mn,Fe)3Si2; the associated Table 2 (actually Table 3) is not mentioned until the description on line 166. It is hard for readers to fully understand what is expressed.

Response to 5: Addressed. The table including the volume fraction measurements has now been referenced at the very start of the microstructural and the figures have been updated.

6. In Table 4 (actually Table 5) on line 266, "A362 As-Cast Component" seems to be wrong.

Response to 6: Addressed.

7. In the article, the description of Figures 7 to 10 refers to "Al15(Fe,Mn)3Si2 phase", but Figures 7 to 10 show "Al(Mn,Fe)Si phase". The figure and text should be revised to be consistent.

Response to 7: Addressed. All figure have been updated to have the same format and clear labelling scheme.

8. The doubt as in question 1 appears again; in Table 5 on line 263 (actually Table 6) and in Table 6 on line 290 (actually Table 7), the item name "Fin" is different with the "Wall" shown in Figure 1. A uniform name should be used for the specified objects throughout the article.

Response to 8: We realize now how that is confusing for the reader. Due to complex curvatures of the wall (AH and TH) section of the gearcase, a tensile sample could not be extracted from this section. Originally we thought that since the wall section experienced that the quickest cooling rate, we extracted a tensile sample from another rapidly solidifying area of the gearcase the “Fin” as a type of reference. The description of the fin section is not required for this paper and thus we have omitted it from the paper.

9. In the tensile test results of the case as-cast (observed in Table 5), the UTS of the other three items except the Fin item are lower than the UTS of MercAlloy™ A362; and the YS of the other three items except the Shaft item are all higher than the YS of MercAlloy™ A362. On the other hand, in the T5 heat-treated tensile test results (observed in Table 6), whether it is UTS or YS, the values ​​of all items (Reference, Shaft, Hub, Fin) are lower than the values ​​of MercAlloy™ A362. As for the reasons for the different results of these two cases, the author should provide more descriptions.

Response to 9: Addressed. The discussion surrounding the fin sample has been removed from the paper. It added too much confusion without adding to the paper. The differences between the Mercalloy™ A362 were presumed to be associated with the difference in casting processes (mentioned in the paper). The gearcase is complex and high pressure die cast, where as the Mercalloy™ A362 data came from an actual tensile sample and thus is surely much more free of defects such as porosity. We originally included the reference values for the published Mercalloy™ A362 results as a comparison but this is only adding confusion. Since the Mercalloy™ A362 sample was not cast in the same way and we haven’t presented the microstructure in this article, we see no benefit for it to be included in the paper. As a result we have omitted it from the paper.

10. The reference number must be revised.

Response to 10: Agreed, that was a formatting issue related to transferring the document to the Metals format.

Reviewer 2 Report

Manuscript Number: Manuscript ID metals-1146792 entitled:

The Effects of Heat Treatment on the Microstructure and Tensile Properties of a HPDC Marine Transmission Gearcase

General comment

Improving the performance and increasing the efficiencies of marine transportation remains actual.

This manuscript reports the development of new alloy, MercAlloy 362™.

A careful grammar and spelling check should be performed. Some recommendations and observation are listed below:

  1. Some phrases should be rewrite or to be reformulate, for a better understanding:

In Abstract

At R10, R263, R272, R286, R290, etc.  Correct with the same notation (MercAlloy 362™  or Mercalloy A362™) 

R10 the phrase “… alloy was designed to remove mass from….” . possible another formulation: (a significant weight reduction).

At R111

“……. and AW specimens are shown in Figure2 to Table5”. Reformulate (Figures 2 to Figure 5).

At R117

“…..This  phase  has  been  observed  in  literature  to  have  various  morphologies  and  compositions depending on the ... “ Rephrase the sentence for clarity. (The literature mentions for…)

Use the same abbreviation for the sample in all determination (AH, AR, AS, AW or Hub, Reference, Shaft, Fin). For tensile properties and microstructural characterization.

  1. Regarding Materials and Methods the samples for microstructural analysis were further prepared? Add all the steps followed for sample preparation (polished, etched …).

  1. Some section showed a directional solidification (multiple dendrites growing in a preferred direction) and the   dendrites   in   the   AR section   present random   orientation.  Can author give more explanation?

  1. Reformulate the phrases

At 217

Another phase that was detected is a primarily needle or rod shape intermetallic. Name this phase.

At R218

“Moreover, the harmful, needle-like Al8FeMg3Si6 phase was also observed in each of the samples but was present in a marginally higher amount.” What means marginally amount?

At R 125.

 “Not only does the morphology lower stress concentrations as compared to the needle-like morphology, but the phase also lowers the interfacial energy with the matrix and as a result improves the alloys toughness “.  It is difficult to follow. Explain.

At R248

“The larger the size of the porosities, the lower the fatigue life of the material due to the ease in which cracks may combine and propagate through a porous metal.”  

  1. the Al15(Mn,Fe)3Si2 has a more favorable geometry that lowers inter-303 facial energy between the precipitate and matrix and generally results in improved 304 tensile performance when replacing the Al8FeMg3Si6 phase.

  1. Compare the microstructure observed in the current work with other / or similar microstructure reported in literature.

  1. Usually the bibliography is not included in the conclusions.

  1. The authors can introduce in the References an earlier work regarding “Residual Stress Analysis of A362 Aluminum Alloy Gear Case Using Neutron Diffraction” Materials Science Forum Vol. 941, pp 1288-1294, doi:10.4028/www.scientific.net/MSF.941.1288, 2018 Trans Tech Publications, Switzerland.

In this paper the conclusion was:

“It is concluded that the heat treatment process extends the lifetime of the component, however, it may not completely eliminate the cracking problem. Farther studies are currently nearing completion, targeting the mass production of the redesigned gearcase.”

The authors can comment and compare the data obtained with those previously published.

Author Response

Reviewer 2

Thank you for your constructive comments and detailed review. We hope the changes we made satisfactorily address your concerns and suggestions.

Improving the performance and increasing the efficiencies of marine transportation remains actual.

This manuscript reports the development of new alloy, MercAlloy 362™.

A careful grammar and spelling check should be performed. Some recommendations and observation are listed below:

  1. Some phrases should be rewrite or to be reformulate, for a better understanding:

In Abstract

At R10, R263, R272, R286, R290, etc.  Correct with the same notation (MercAlloy 362™  or Mercalloy A362™) 

Response to 1: Addressed throughout the paper.

R10 the phrase “… alloy was designed to remove mass from….” . possible another formulation: (a significant weight reduction).

Response to 1: Addressed, changed “remove mass from” to “lighten”.

At R111

“……. and AW specimens are shown in Figure2 to Table5”. Reformulate (Figures 2 to Figure 5).

Response to 1: Addressed.

At R117

“…..This  phase  has  been  observed  in  literature  to  have  various  morphologies  and  compositions depending on the ... “ Rephrase the sentence for clarity. (The literature mentions for…)

Response to 1: Addressed.

Use the same abbreviation for the sample in all determination (AH, AR, AS, AW or Hub, Reference, Shaft, Fin). For tensile properties and microstructural characterization.

2. Regarding Materials and Methods the samples for microstructural analysis were further prepared? Add all the steps followed for sample preparation (polished, etched …).

 Response to 2: Addressed. Full microstructural preparation now included in the paper.

3. Some section showed a directional solidification (multiple dendrites growing in a preferred direction) and the   dendrites   in   the   AR section   present random   orientation.  Can author give more explanation?

Response to 3: Addressed. The authors have included additional information about the micrographs. The micrographs shown in the paper were selected to purposely show the phases present in the alloy and thus the bulk of the material at each location may differ slightly. Texture plays a large role in neutron diffraction experiments (previous study) and the authors confirmed that texture was not largely present in the material.

4. Reformulate the phrases

At 217

Another phase that was detected is a primarily needle or rod shape intermetallic. Name this phase.

Response to 4: Addressed. That entire sentence should not have been there and has been removed.

At R218

“Moreover, the harmful, needle-like Al8FeMg3Si6 phase was also observed in each of the samples but was present in a marginally higher amount.” What means marginally amount?

Response to 4: Addressed. Changed to “slightly”

At R 125.

 “Not only does the morphology lower stress concentrations as compared to the needle-like morphology, but the phase also lowers the interfacial energy with the matrix and as a result improves the alloys toughness “.  It is difficult to follow. Explain.

Response to 4: Addressed, reworded the phrase.

At R248

“The larger the size of the porosities, the lower the fatigue life of the material due to the ease in which cracks may combine and propagate through a porous metal.”  

 Response to 4: Addressed, reworded the phrase.

5. the Al15(Mn,Fe)3Si2 has a more favorable geometry that lowers inter-303 facial energy between the precipitate and matrix and generally results in improved 304 tensile performance when replacing the Al8FeMg3Si6 phase.

  Response to 5: Addressed, reworded the phrase.

6. Compare the microstructure observed in the current work with other / or similar microstructure reported in literature.

Response to 6: Addressed, additional discussion and references have been added to the document

7. Usually the bibliography is not included in the conclusions.

 Response to 7: The bibliography is included under a separate section labelled as “References” which is located after the conclusions.

8. The authors can introduce in the References an earlier work regarding “Residual Stress Analysis of A362 Aluminum Alloy Gear Case Using Neutron Diffraction” Materials Science Forum Vol. 941, pp 1288-1294, doi:10.4028/www.scientific.net/MSF.941.1288, 2018 Trans Tech Publications, Switzerland.

In this paper the conclusion was:

“It is concluded that the heat treatment process extends the lifetime of the component, however, it may not completely eliminate the cracking problem. Farther studies are currently nearing completion, targeting the mass production of the redesigned gearcase.”

The authors can comment and compare the data obtained with those previously published.

 Response to 8: The results from the previous publications have been mentioned throughout the present article. The previous publications focused primarily only on the evolution of residual stress as a function of heat treatment or machining. Those publications indicated that the T5 heat treatment successfully alleviated a considerable portion of the residual stress, but it was unsure how the T5 heat treatment affected the microstructure. The current article explains the effects of the T5 treatment, specifically that the T5 heat treatment did not lead to any major changes in the microstructure and instead the stress relief described in the previous publications was more likely associated with plastic deformation caused by a the temperature sensitive yield strength being surpassed during the heat treatment process.

Reviewer 3 Report

Dear Authors,

Congratulations on your work. However, you can improve your paper if you are able to properly address the following comments:

  1. The Introduction seems like a contextualization of the problem, not a real state-of-the-art regarding this kind of alloys and the root-causes behind fracture events. Thus, to reformulate the entire Introduction is mandatory. The contextualization of the problem should apear at the end of the Introduction, or in the beginning of the Methods.
  2. In the Methods, please describe how the samples have been prepared.
  3. Because the level of stress is a key factor to understand the problem, and it is essential to understand the effect of each condition, I cannot understand because XRD analysis has not been considered to measure the level of stress in each area and material state.
  4. In table 2, it is not clear how the values have been obtained. Please clarify.
  5. There are some formatting concerns, like in Figure 6. Please correct.
  6. The discussion is poor and is mixed with the results. Please insert a new section just devoted to DISCUSSION, and include more references to support it.

Good luck.

Kind regards,

Reviewer

Author Response

Reviewer 3

Dear Authors,

Congratulations on your work. However, you can improve your paper if you are able to properly address the following comments: Thank you!

1. The Introduction seems like a contextualization of the problem, not a real state-of-the-art regarding this kind of alloys and the root-causes behind fracture events. Thus, to reformulate the entire Introduction is mandatory. The contextualization of the problem should apear at the end of the Introduction, or in the beginning of the Methods.

Response to 1: Addressed. The introduction has been updated to include additional background information.

2. In the Methods, please describe how the samples have been prepared.

Response to 2: Addressed, the sample preparation has been included.

3. Because the level of stress is a key factor to understand the problem, and it is essential to understand the effect of each condition, I cannot understand because XRD analysis has not been considered to measure the level of stress in each area and material state.

Respond to 3: The residual stress was actually obtained using neutron diffraction in a previous study, as referenced. The previous residual stress study found that the T5 heat treatment lowered the magnitude of stress present in the gearcase but the reasons were not well understood. The present study confirms that the alleviation of stress is not associated with microstructural changes and is more likely associated with the magnitude of residual stress surpassing the temperature-sensitive yield strength of the alloy during heat treatment, leading to plastic deformation and stress release.

4. In table 2, it is not clear how the values have been obtained. Please clarify.

Response to 4: The paragraph before Table 2 indicates that the procedures outlined in ASTM standard E562-02 were used to determine the volume fraction of the intermetallics. Additional discussion was also added to the document.

5. There are some formatting concerns, like in Figure 6. Please correct.

Response to 5: The paper has been reviewed once more and the formatting has been checked and fixed where needed. We believe the odd formatting for figure 6 was due to changing the document from a regular word format to the Metals specific format.

6. The discussion is poor and is mixed with the results. Please insert a new section just devoted to DISCUSSION, and include more references to support it.

Response to 6: The discussion has been improved and the journal guidelines indicate that the discussion portion may be included into the results sections:

Discussion: Authors should discuss the results and how they can be interpreted in perspective of previous studies and of the working hypotheses. The findings and their implications should be discussed in the broadest context possible and limitations of the work highlighted. Future research directions may also be mentioned. This section may be combined with Results.

Reviewer 4 Report

Manuscript under review provides a wide experimental investigation concerning the effects of heat treatment on the Microstructure and Tensile Properties of a  Marine Transmission Gearcase produced by HPDC.

Microstructural characterization is complete with very interesting findings which are well documented in the text and the associated micrographs.

A recommendation to the authors is to discuss the low values of the elongation (Tables 5 & 6) and correlate them to the microstructure.

Author Response

Reviewer 4

Manuscript under review provides a wide experimental investigation concerning the effects of heat treatment on the Microstructure and Tensile Properties of a  Marine Transmission Gearcase produced by HPDC.

Microstructural characterization is complete with very interesting findings which are well documented in the text and the associated micrographs. Thank you!

1. A recommendation to the authors is to discuss the low values of the elongation (Tables 5 & 6) and correlate them to the microstructure.

Response to 1: Absolutely. The low magnitudes of elongation are due to the relatively high amount of porosity in the gearcase. The porosity is caused by turbulent flow and acts as a crack initiation site and facilitates crack propagation, thus why we observed low elongation. This has been addressed in the paper.

Round 2

Reviewer 3 Report

Thank you for properly addressing my comments and suggestions.

Kind regards,

Reviewer